# Exploitation of Agro-Industrial Waste as Potential Source of Bioactive Compounds for Aquaculture

**DOI:** 10.3390/foods9070843

**Published:** 2020-06-28

**Authors:** Nayely Leyva-López, Cynthia E. Lizárraga-Velázquez, Crisantema Hernández, Erika Y. Sánchez-Gutiérrez

**Affiliations:** 1Cátedras CONACYT-Centro de Investigación en Alimentación y Desarrollo, A.C. (Food and Development Research Center), Unidad Mazatlán. Av. Sábalo Cerritos S/N, Mazatlán 82112, Sinaloa, Mexico; nayely.leyva@ciad.mx; 2Centro de Investigación en Alimentación y Desarrollo, A.C. (Food and Development Research Center), Unidad Mazatlán. Av. Sábalo Cerritos S/N, Mazatlán 82112, Sinaloa, Mexico; cesmeralda.lizarraga@gmail.com (C.E.L.-V.); erika.sanchez@ciad.mx (E.Y.S.-G.)

**Keywords:** food by-products, biological activities, phenolic compounds, antioxidant, immunostimulants, prebiotics, fish

## Abstract

The agroindustry generates a large amount of waste. In postharvest, food losses can reach up to 50%. This waste represents a source of contamination of soil, air, and bodies of water. This represents a problem for the environment as well as for public health. However, this waste is an important source of bioactive compounds, such as phenolic compounds, terpenes, and β-glucans, among others. Several biological activities have been attributed to these compounds; for example, antioxidant, antimicrobial, gut microbiota, and immune system modulators. These properties have been associated with improvements in health. Recently, the approach of using these bioactive compounds as food additives for aquaculture have been addressed, where it is sought that organisms, in addition to growing, preserve their health and become disease resistant. The exploitation of agro-industrial waste as a source of bioactive compounds for aquaculture has a triple objective—to provide added value to production chains, reduce pollution, and improve the well-being of organisms through nutrition. However, to make use of the waste, it is necessary to revalue them, mainly by determining their biological effects in aquaculture organisms. The composition of bioactive compounds of agro-industrial wastes, their biological properties, and their application in aquaculture will be addressed here.

## 1. Introduction

Population growth and urbanization have increased the demand for processed foods. This has led to the development of food industries to meet the needs of consumers. In particular, agricultural industries generate a large amount of waste during the collection, storage, transport, and processing of raw materials [1]. This represents an environmental pollution problem due to the waste being mainly made up of organic matter. Organic matter represents a source of bioactive compounds. A bioactive compound is a substance that has a biological activity. In a broader sense, it is a substance that has an effect or can trigger a physiological response on a living organism. The effect may be negative or positive depending of the chemical structure, the dose, and the bioavailability of the substance [2]. However, bioactive compounds are widely recognized for promoting health benefits [3].

In Mexico, the processing industries of fruits (lemon, pears, tomato, apples, papaya, pineapple, banana, and oranges), cereals (corn), and vegetables (beans, cabbage, carrots, lettuce, and potatoes) generate around 76 million tons annually of waste [4]. The husks, bark, seeds, and pomace represent the largest amount of waste, which are made up of a wide variety of bioactive compounds with multiple biological properties such as antioxidant, immunostimulant, antimicrobial, anticancer and prebiotic.

The exploitation of waste for the development of food products with added value could make possible the generation of additional profits. The agricultural industry, in collaboration with the scientific community, has directed efforts toward the design of appropriate methods for the extraction and purification of bioactive compounds from waste for the development of functional food. It is currently known that the cost of technologies for the purification of bioactive compounds exceed reprocessing costs, so the full use of waste with functional properties as additives could lead the food industry to reduce residues and increase its profitability [5].

Currently, the aquaculture sector is in need to find low-cost natural compounds as food additives, which might promote well-being and preserve the nutritional quality of farming organisms, without compromising the environment and the health of consumers. Therefore, the use of plant residues as additives represents a promising alternative. However, these residues require appropriate characterization, since some studies report that the use of phytochemicals might exert anti-nutritional effects [6,7].

## 2. Bioactive Compounds from Agro-Industrial Waste

There is a wide variety of bioactive compounds found in residues derived from the cultivation and processing of agricultural products. Any part of plants, such as husk, seeds, leaves, roots, and stems, can be considered as a source of bioactive compounds [8]. Some of the most important bioactive groups are briefly described next.

### 2.1. Phenolic Compounds

Phenolic compounds (PCs) are secondary metabolites of plants that are usually esterified or glycosylated [9] and are mainly composed of an aromatic ring (hydrophobic domain) and one or more hydroxyl groups (hydrophilic domain) attached to it. Within this group of compounds are phenolic acids (hydroxycinnamic and hydroxybenzoic), flavonoids, coumarins, xanthones, chalcones, stilbenes, lignins, and lignans (Figure 1a) [10].

Phenolic compounds are known for having several biological properties, such as antioxidants [11], immunostimulants [12], and microbiota modulators [13]. Besides, they are also recognized for their antibacterial [14], antiparasitic [15], antiviral [16], anti-inflammatory [17], anticancer [18], and antihypertensives [19] effects. The biological properties of PCs, especially the antioxidant, are related to their chemical-structural characteristics. The number and position of hydroxyl groups, the presence of double bonds, and the ability to delocalize electrons determine the ability of PCs to scavenge free radicals and donate hydrogen atoms [20]. Other mechanisms by which PCs can exert their biological activity is by interacting with components of the cell membrane, enzymes, and transcription factors, as well as receptors [21]. For example, PCs provide antioxidant protection to membrane through the interaction of their hydrophilic and hydrophobic domains with the polar head and non-polar chain of lipids bilayer [21]. Flavonoids exert antiallergenic and anticarcinogenic activities by interacting with membrane raft-associated proteins [22] and decrease superoxide anion production in vascular cells through the inhibition of the translocation of p47^phox^ nicotinamide adenine dinucleotide phosphate (NADPH) oxidase subunit in the endothelial cell membrane [19]. Moreover, PCs exert indirect antioxidant and anti-inflammatory activities by activation of Nrf2 and inhibition of NF-κB translocation into the nucleus, respectively [23].

### 2.2. Terpenes

Terpenes are compounds formed by isoprene units (C_5_H_8_). These compounds are classified according to the number of isoprene units condensed [24]. Within the group of terpenes are essential oils and carotenoids, which are characterized by their antioxidant activity. Also, essential oils are known to exert microbicidal effects [25,26].

Terpenes biological activities are related to their chemical structure (Figure 1b). For instance, the antioxidant activity of essential oils is mainly due to the presence of phenolic type components, such as thymol and carvacrol; therefore, the antioxidant action mechanisms of these terpenes are similar to those previously mentioned for PCs [27]. On the other hand, the antimicrobial effect of essential oils is related to their hydrophobicity. This property allows essential oils to cross the cell wall and cytoplasmic membrane and disorganize the structure of its components. Besides, essential oils are capable of inhibiting enzymes of energy regulation and synthesis of structural components [28].

### 2.3. Dietary Fiber (β-glucans)

Dietary fiber is composed of polymers of three or more carbohydrate units that are resistant to the activity of endogenous digestive enzymes, and therefore cannot be hydrolyzed or absorbed by the small intestine [29]. Fiber is classified as insoluble and soluble fiber. Soluble fibers, such as β-glucans, fructooligosaccharides, galactooligosaccharides, and some pectins, are fermented by the intestinal microbiota and give rise to short-chain fatty acids (acetate, propionate, and butyrate) [30].

Particularly in aquaculture, β-glucans (Figure 1c) are recognized for their immunostimulatory activity [31]. These compounds are polysaccharides made up of glucose units linked by glycosidic bonds. These are found as components of the cell wall of plants and yeasts mainly, but also in some species of algae and fungi. The most relevant β-glucans are β-1,3 and β-1,6 [32,33]. The immunostimulatory activity of β-glucans depends on their recognition and binding to membrane receptors (for example, dectin-1, and CR3). Besides, the degree of polymerization, the degree and type of branching, and the structural conformation of β-glucans affect their interaction with receptors [34].

### 2.4. Glucosinolates

The glucosinolates are glycosides formed by a β-D-glucopyranose residue linked to a (Z)-N-hydroximinosulfate ester by sulfur bridges, and an amino acid derivative radical. These compounds are found in all species belonging to the *Brassica* family, such as canola, broccoli, arugula, and mustard [35]. Glucosinolates can be classified based on their amino acid precursor into aliphatic, aromatic, and indole [36,37].

Glucosinolates and the products derived from their degradation (isothiocyanates) show antioxidant, anticancer and antibacterial activity. These compounds act as indirect antioxidants because they are capable of modulating the activity of xenobiotic-metabolizing enzymes (Phase I and Phase II), which triggers the long-lasting antioxidant reactions [38]. On the other hand, the bactericidal activity of the products of the metabolism of glucosinolates has been related to the inhibition of intracellular enzymes responsible for ATP synthesis in pathogenic bacteria [39,40].

### 2.5. Saponins

Saponins are amphipathic molecules composed of sugar residues linked to a system of polycyclic rings (sterols and triterpenes) through glycosidic bonds [41]. These compounds are present in plant products, such as agave or legumes [42,43]. 

Saponins have immunostimulatory effects [44]. The structural characteristic associated with this activity is the presence of an aldehyde group at position C19 and C4 of the aglycone [45]. Besides, saponins exert microbiota modulating effect, which is related to their antimicrobial activity. Furthermore, saponins can dissociate the cell membrane, and therefore, the flow of extracellular and intracellular components is enabled [46]. The effectiveness of saponins is enhanced against Gram-positive bacteria, while Gram-negative bacteria are more resistant, possibly due to the presence of the double lipid membrane in the latter [47].

Despite the beneficial properties attributed to bioactive compounds, they might possess anti-nutritional effects due to inhibition of the digestive protease activity and formation of complexes with proteins [48,49]. Since bioactive compounds might exert beneficial effects on organisms of importance for aquaculture, their use as food additives has been explored. Nevertheless, the effect of these compounds on the metabolism and growth of species is still to be understood.

## 3. Biological Properties and Mode of Action of Bioactive Compounds

### 3.1. Antioxidant Activity

Free radicals are atoms or molecules that have a missing electron in the last orbital, which gives them instability and high reactivity. Free radicals reach balance by receiving electrons from other molecules, such as carbohydrates, proteins, lipids, and nucleic acids [50]. These reactive molecules are produced during normal cellular metabolism, some examples are superoxide anion (O^2−^), hydroxyl radical (OH^−^) and hydroperoxyl radical (HO^2−^) [51]. An excess in the levels of free radicals can start harmful effects on important macromolecules, like lipids, proteins and nucleic acids [52]. The lipid peroxidation is caused by free radicals. This process increases the production of free radicals and leads to the formation of aldehydes such as malondyaldehyde (MDA) and 4-hydroxy-2-nonenal (HNE) (Figure 2a), which are characterized by their cytotoxic and mutagenic effects [52,53]. Lipid peroxidation and other cell damages can be prevented with antioxidants. 

Antioxidants are substances capable of neutralizing or reducing the deterioration caused by free radicals [54]. The antioxidant activity can be exerted by directly donating electrons to free radicals to stabilize them or regulating the activity of transcription factors, such as the nuclear factor enhancing the kappa light chains of activated B cells (NF-κB) and the nuclear factor derived from erythroid 2 (Nrf2). These factors participate in the regulation of gene expression of detoxifying and antioxidant enzymes such as superoxide dismutase (SOD), catalase (CAT), and glutathione peroxidase (GPx) (Figure 2b) [55,56]. 

Bioactive compounds act as both direct and indirect antioxidants due to the presence of OH groups, double bond, carbonyl groups, and aromatic rings in their structures. These compounds can regulate the Nrf2 activation by Keap-1-dependent and Keap-1-independent pathways [57]. Keap-1-dependent requires strong electrophiles disrupting the Keap1-Nrf2 complex. Keap1-independent involves the phosphorylation of Nrf2 by protein kinases (ERK, JNK, PKC, p38 MAPK, and Akt) [57]. Both pathways promote the Nrf2 translocation to nucleus and the binding of this factor to sMaf protein and the antioxidant response element (ARE). This initiates the transcription of antioxidant enzymes (Figure 2c), such as SOD, CAT and GPx. Gallic and caffeic acids activate Nrf2 by induction of the post-translational phosphorylation of ERK [22,57], while β-glucans do it via p38 MAPK signaling [58]. Electron rich flavonoids such as epigallocatechin gallate, quercetin, and morin have the ability to form stabilized electrophiles and act as Michael reaction acceptors and, thus, can modify cysteine residues of Keap1 [59]. Carotenoids such as lycopene and adonixanthin can also activate Nrf2, although the activation pathway is still unknown [60].

The NF-κB activation occurs by phosphorylation of protein kinases (PI3K, PKC, JNK, and ERK), which induce phosphorylation of IKKα/β and leads to IκBα phosphorylation. Afterward, the NF-κB is translocated to the nucleus for the transcriptional regulation of antioxidant genes (Figure 2d) [61]. The high electrophilicity of Michael acceptors of PCs also allows to modify IKK cysteine residues and hence, the nuclear translocation of NF-κB [61,62]. However, PCs also can act as anti-inflammatory agents and avoid the NF-κB activation by inhibition of the phosphorylation of IκB-α, ERK, JNK, p38, and MAPK [62]. In this sense, lycopene also can inactivate the NF-κB translocation [63].

The gene expression and activity of antioxidants enzymes (SOD, CAT, GPx) and the levels of lipid peroxidation mediators—namely, malonaldehyde (MDA) are used as markers to determine if a food additive can exert a stimulating action on the antioxidant system and reduce oxidative stress in the organisms [64,65].

### 3.2. Immunostimulant Activity

Immunostimulants, also known as adjuvants or immunomodulators, are compounds or substances that promote the response of the immune system, which might generate resistance of the organisms to the presence of pathogens [66]. Immunostimulants can be obtained from animal, vegetable, and bacterial sources, as wells as from algae, nutritional factors and hormones, or cytokines [67]. Generally evaluated biomarkers used to define the immunostimulant effect of a substance are:(1)The increase in the enzymatic activity of lysozyme and myeloperoxidase (MPO). Lysozyme exerts its microbicidal action by lysis of peptidoglycans, components of the cell wall of Gram-positive bacteria [68], while MPO catalyzes the formation of hypochlorous, hypobromous and hypothiocyanite acids [69].(2)Increase in respiratory burst. When phagocytic cells, such as neutrophils and macrophages, respond to the presence of a pathogen, they trigger the action of NADPH oxidase. This generates superoxide anion (O^2−^). The measurement of this radical by the nitroblue tetrazolium (NBT) reduction method has been considered as an indicator of the phagocytic capacity of the cells of the immune system [70].(3)Increase in the number of red and white cells. The cell count is a measure used to evaluate the effect of some possible immunostimulants on the health of organisms. A reduction in the count of red cells (erythrocytes) implies that the substance is exerting collateral damage (anemia) in the body. An increase in the number of white cells (leukocytes) indicates a greater response of the immune system to a possible infectious agent. Other blood cell indicators are neutrophils count, hematocrit, level of hemoglobin, etc. [66].(4)Other immunological parameters evaluated are complement components, such as soluble proteins, enzymes, and receptors that act in signaling processes, opsonization of pathogenic microbes, phagocytosis and microbial destruction [71]. The concentration of immunoglobulins (Ig) and the level of protein are also frequently evaluated as immunological parameters [66]. Melanomacrophage centers (MMCs), pigmented phagocytic cells (melanin) that act as a rapid response to the presence of an infection, and cytokine levels, such as interleukin-1 (IL-1), IL-6, and interferon-gamma (IFN-γ) are also considered markers of the immune response in fish [72].

The NF-κB factor plays an extremely important role in the immune response of organisms. It is responsible for the expression of inflammatory genes, namely cytokines and enzymes (nitric oxide synthase, NOS). Furthermore, NF-κB is activated under stress conditions, such as oxidative agents or pathogens presence (Figure 2d). These stressors provoke the phosphorylation of IκB accompanied by the ubiquitination and degradation of the protein. The IκB degradation releases the NF-κB and allows it to enter the nucleus and activate the expression of immune-related genes [73]. Therefore, the NF-κB modulation is a key factor to evaluate the immunomodulatory effect of bioactive compounds. 

It has been proposed that PCs, such as flavonoids, could exert anti-inflammatory effects by inhibiting the action of the NF-κB factor. For instance, the -OH groups attached to the C3′ and C4′ in B ring present in (−)-epicatechin may interact with the nuclear fractions p50 and Rel A of NF-κB, which prevents the binding of the factor with the specific DNA-kB sites, hence blocking the expression of cytokine-related genes. [21]. In contrast, it has been demonstrated that quercetin, a flavonoid, increases IFN-γ secretion while reduces IL-4 levels in blood mononuclear cells [74]. An up-regulation of IFN-γ has been associated to a better adaptive immune response in fish [75]. Due to these inconsistencies, more research is needed on bioactive compounds to obtain conclusive data. Monoterpenes, such as limonene, have been reported to inhibit the phosphorylation of IkB and therefore block the translocation of the factor NF-κB to the nucleus [76]. On the other hand, β-glucans exert their immunomodulatory effect by promoting the activation of NF-κB via recognition and binding by the receptors dectin-1 and CR3, among others. Dectin-1 recognizes glucans with both β-(1–3) and β-(1–6) bonds, therefore the linking strength depends on the size of the molecule and branching degree. Through this recognition, the NF-κB is activated to induce cytokine and chemokine synthesis [34]. In fish, a member of the C-type lectin receptor, different from dectin-1, might be responsible of the β-glucan recognition. Two C-type lectin domain-encoding genes, named *clec4c* and *sclra*, were identified in primary macrophages of common carp (*Cyprinus carpio* L.) [77].

### 3.3. Intestinal Microbiota Modulation

The modulation of the intestinal microbiota or the induction of changes in the composition of the host’s microbiota is achieved through the use of probiotics, prebiotics, and synbiotics [78]. A prebiotic is defined as a non-digestible compound that, through its metabolism by microorganisms, modulates the composition or activity of the intestinal microbiota, and confers a beneficial physiological effect on the host [79]. Generally, the modulation of the microbiota is carried out for the benefit of the host to increase the abundance of beneficial bacteria and inhibit the growth of pathogenic bacteria. The latter can be achieved by selecting additives and/or functional ingredients, that once incorporated into the food might exert the activity. 

The immune modulatory effect of β-glucans is strongly related to their chemical structure and depends on the type of branching, the degree of ramifications, solubility, molecular weight, tertiary structure, polymer charge, and solution conformation (triple or single helix or random coil) [80]. For instance, soluble β-glucans are fermented by the gut microbiota in the large intestine, such as populations of bifidobacteria and lactobacilli, which produce cell-associated glycosidases. After fermentation, β-glucans are metabolized to produce short-chain fatty acids, such as acetic, propionic and butyric acids [81]. These short-chain fatty acids have been reported to possess biological activity, such as reducing cholesterol levels in humans [82]. The exact molecular mechanism by which β-glucans affect gut microbiota is still not clear [83]; therefore, more research is needed in this topic.

Dietary fibers are the plant bioactive compounds more studied due to their ability to modify the intestinal microbiota (prebiotic effect) [79]. Some PCs that are not absorbed in the intestine might be metabolized by intestinal bacteria, acting as modulators of the microbiota [84]. The precise mechanism by which PCs exert their microbiota modulation effect still remains to be elucidated. Nevertheless, there are reports that show that gut microbiota transforms PCs that are linked to glycosides [85]. This transformation occurs through a series of reactions of hydroxylation, methylation, dehydrogenation, isomerization, glycosylation, etc. [13]. For instance, epigallocatechin gallate, from tea, was converted to its derivates gallic acid and epigallocatechin after 36 h of anaerobic fermentation in vitro inoculated with fecal slurry from healthy women. Furthermore, PCs from green tea, oolong tea, and black tea significantly increased population of bacteria belonging to genera *Bifidobacterium* and *Lactobacillus-Enterococcus* spp. and suppressed the growth of *Bacteroides-Prevotella* and *Clostridium histolyticum* [86].

The modulatory effect of bacteria by terpenes has been widely associated to the antibacterial properties of these bioactive compounds. Terpenes can pass through the cell membrane of bacteria and cellular organelles due to their hydrophobic properties and therefore disarrange the structure of the phospholipidic bilayer and increase permeability. This causes the leakage of relevant molecules and ions in the bacteria [26]. Furthermore, by destabilizing the membrane structure of cellular organelles, such as mitochondria and endoplasmic reticulum, terpenes inhibit enzymatic reactions responsible of the energy metabolism, as well as the synthesis of structural macromolecules. Besides, it is proposed that terpenes, namely essential oils, exert a higher antibacterial effect on Gram-positive than in Gram-negative bacteria. This might be due to the fact that Gram-negative bacteria possess an additional structure (cell wall) that shows hydrophilic properties, which might block the pass of the hydrophobic structure of the terpenes [87].

## 4. Use of Bioactive Compounds from Agro-Industrial Waste in Aquaculture

Aquaculture is an economic sector that shows a broad growth. According to data from the Food and Agriculture Organization of the United Nations [88], aquaculture contributes around half of the production of fish destined for human consumption. Due to the accelerated growth and high demand for aquaculture products, farming has intensified; that is, a greater number of organisms are produced in smaller spaces. Furthermore, other farming factors, such as poor diet, poor water quality, and changes in temperature and pH, might cause stress, suppress the immune system of organisms, and negatively affect their health condition. These conditions might increase the apparition and rapid spread of infectious diseases, which are a major problem for the aquaculture industry due to the economic losses. Traditionally, antibiotics are used to mitigate this problem; however, their unselective use has turned out to be a dangerous solution, due to the appearance of antibiotic-resistant bacteria, as well as the use of these chemicals is undesirable for the final consumer.

Therefore, there is a need to seek alternative options to reduce the problems of disease occurrences by increasing the antioxidant and immune responses in organisms. Recently, there has been a special interest in bioactive compounds, since they have been shown to have multiple properties, such as to promote growth and improve the health of aquatic organisms by reducing oxidative stress and stimulating the immune system, which provides resistance to diseases (Figure 3).

### 4.1. Bioactive Compounds as Antioxidants in Aquaculture

In aquaculture, there is enough information on the use of bioactive compounds from medicinal plants, as food additives, to increase the antioxidant response and counteract the effects of oxidative stress (lipid oxidation and loss of nutritional quality). In this regard, bioactive compounds from plant residues have been poorly explored.

Corn, rice, wheat, and sorghum are among the most consumed cereals in the world. From the cereal processing a significant amount of residue is generated. This waste could be used for the development of functional foods with antioxidant activity. In this context, some studies have been carried out to evaluate the efficacy of corn and sorghum residues as antioxidant additives in fish feed. Catap et al. [89] reported that the dietary administration of corn silk extract (*Zea mays*) lowered the level of lipid peroxidation in the liver of Nile tilapia (*Oreochromis niloticus*) under paracetamol-induced oxidative stress. Corn silk is an important source of flavonoids such as luteolin, formononetin, maizine, and apigenin [90]. In general, these compounds are recognized for neutralizing reactive oxygen species (ROS) and modulating antioxidant enzyme activities [91]. Therefore, this study might suggest that flavonoids improve the antioxidant response in Nile tilapia liver under induced stress. On the other hand, Lee et al. [92] indicated that the dietary inclusion of residues from the distillation of sorghum (200 g/kg), increases the antioxidant activity and delays the oxidation process of low-density lipoproteins in the plasma of the Lysa (*Mugil cephalus*). Among the main bioactive compounds present in sorghum are phenolic acids (caffeic, ferulic and chlorogenic acids) and flavonoids (apigeninidin, luteolinidin, and naringenin), which have been directly related to its antioxidant activity [93]. Therefore, the efficacy of sorghum to increase antioxidant activity and delay the oxidation of low-density lipoproteins in the plasma of *M. cephalus* is attributed to the ability of PCs to neutralize free radicals.

Waste from fruit processing has received little attention in the aquaculture area, even though it is known to be one of the main sources of bioactive compounds. There are currently few studies related to the use of fruit residues. For example, Giri et al. [94] evaluated the effect of the dietary inclusion of banana peel (*Musa acuminata*) (10, 30, 50 and 70 g/kg) at different feeding times (30 and 60 days) on the formation of MDA and the activity of SOD, GPx, and CAT in rohu (*Labeo rohita*) liver, infected with *Aeromonas hydrophila*. Fish feed including banana peel (50 g/kg and 70 g/kg) showed a significant decrease in MDA levels in both feeding times. Superoxide dismutase and CAT activity increased in fish fed 50 g/kg of banana peel during 60 days of feeding, while GPx activity showed an increase after 30 days of feeding fish the banana peel (30, 50, and 70 g/kg). The authors suggested that the diversity of bioactive compounds identified in the banana peel, such as phenolic acids, flavonoids, and carotenoids, is responsible for improving the hepatic antioxidant response in rohu. Furthermore, Vicente et al. [95] evaluated the effect of orange peel fragment (OPF), as a food additive, on the antioxidant enzyme activity of Nile tilapia subjected to heat/dissolved oxygen-induced stress. In the study, fish were fed diets with different inclusion of OPF (0, 0.2, 0.4, 0.6, and 0.8%) for 70 days. At the end of the feeding trial, fish were subjected to stress conditions (32 °C/2.3 mg/L dissolved oxygen) for three days. Before stress, SOD, CAT, and GPx activities were higher in the non-supplemented group. Nevertheless, after stress, OPF supplementation increased SOD, CAT, and GPx activities. The increase in the antioxidant enzyme activities in Nile tilapia liver could be associated with the presence of hesperidin, a flavonoid, in the orange peel fragments [96]. This flavonoid upregulates the Nrf2 gene expression which improves antioxidant enzyme activity, therefore minimizing oxidative stress [97]. Additionally, Lizárraga-Velázquez et al. [98] reported that supplemented diets with 50 mg and 100 mg of PCs from mango peel extract (MPE) per kg feed decreased MDA levels as a measure of the lipid peroxidation in zebrafish (*Danio rerio*) muscle. Moreover, the authors indicated that the dietary administration of 150 mg and 200 mg PCs from MPE per kg feed increased the hepatic CAT activity without affecting the growth and feed utilization of zebrafish. Antioxidant effects were attributed to the presence of gallic, 2-hydroxycinnamic and protocatechuic acids, mangiferin, quercetin, methyl gallate, and ethyl gallate in the MPE. These latest studies merit further investigation to validate the use of phenolic compounds as feed supplements.

Beta-glucans from mushroom (*Pleurotus pulmonarius*) stalk waste (MSW) have also been explored as antioxidants in aquaculture. For instance, Ahmed et al. [99] evaluated the use of hot water extracts (HWE) from MSW as an additive in fish feed and determined the effect on growth performance and the in vivo antioxidant status of Nile tilapia. When the HWE from MSW, rich in β-glucan content (20.05 ± 0.44%), was added to the diet (10 g/kg), SOD and CAT activities in the liver and kidney were enhanced. The authors mentioned that the effect of β-glucan on SOD and CAT activity might help to prevent the deleterious effects of ROS on organisms. Furthermore, the protection of β-glucans present in HWE of MSW against oxidative stress caused by pH fluctuations in Nile tilapia was also evaluated [100]. Administration of 5 g/kg and 10 g/kg under pre-stress conditions increased SOD and CAT activities in the liver and kidney respectively. Nevertheless, the activity of these enzymes was reduced in liver and kidney samples due to pH changes (5.5 and 10.5). The authors concluded that the supplementation of β-glucans from MSW in the diet for tilapia, enhanced the antioxidant enzyme activities in vivo, which led fish to reduce stress for pH fluctuations and therefore show a normal growth.

### 4.2. Bioactive Compounds as Modulators of the Immune System and Resistance to Infections

The use of immunostimulants to control aquaculture diseases emerges as an alternative to the use of antibiotics. This topic is gaining particular interest in the scientific community and different sources of obtaining these bioactive have been proposed, such as medicinal plants [101]. However, the use of agro-industrial waste has been less explored, and therefore, the literature in this regard is scarce. 

Phenolic compounds are a group of phytochemicals that have been studied in aquaculture as food additives because of their potential as immunostimulants. Particular interest has been taken in PCs from grape seed, a material that is discarded from wine processing. In this regard, Magrone, Fontana, Laforgia, Dragone, Jirillo and Passantino [75] evaluated the effect of grape seed extracts (Canosina Nero di Troia *Vitis vinifera*) on the immune response of juveniles of *Dicentrarchus labrax* L. In this study, the authors demonstrated that the incorporation of 0.1 and 0.2 g/kg of phenolic extract in feed for *D. labrax*, reduces the levels of IL-1β and IL-6 in the intestine, while the concentration of IFN-γ in the spleen increased. The effect of PCs on the levels of cytokines could be due to the modulation that they exert on the NF-κB factor, which regulates the expression of several cytokines [102]. In addition, the number of MMCs increased. These results show that the diet with polyphenols reduces intestinal inflammation by reducing the levels of proinflammatory cytokines, while the increase in interferon expresses a more robust adaptive immune response. Another aspect is that the increase in the number of MMCs, which contain melanin, is associated with protective functions against pathogens [72]. Furthermore, Arciuli et al. [103] evaluated the effect of PCs extracted from grape seeds on the activity of MMCs, dopa-oxidase, and peroxidase in commercial size fish of *D. labrax*. The administration of 0.2 g/kg in the diet of *D. labrax* increased the activity of dopa-oxidase and peroxidase. These enzymes participate in the synthesis of melanin. The presence of the latter in fish is associated with protective functions against pathological or stress conditions [72]. From these studies, it can be concluded that the addition of PCs from grape seeds increases melanin levels in fish. As previously mentioned, the presence of this pigment is associated with the resistance of organisms against pathogens. Hence PCs could be an option to improve the health status of fish. However, more research is required to validate this effect through a challenge of organisms fed bioactive compounds in the presence of pathogens of interest.

Hoseinifar et al. [104] indicated that the dietary administration of olive waste cake (OWC) (0.5, 2.5, and 5.0 g/kg of feed), increased weight gain (WG) and specific growth rate (SGR), and decreased the feed conversion ratio (FCR) in rainbow trout (*Oncorhynchus mykiss*). The authors also reported that the dietary inclusion of 2.5 g/kg and 5.0 g/kg OWC feed increased total Ig concentration and mucosal lysozyme activity, also up-regulated relative expression of gut IL-8 gene. While, the supplementation of 2.5 g/kg OWC increased the serum lysozyme activity. The immunomodulatory effects of OWC are related to the presence of PCs (hydroxytyrosol, tyrosol, caffeic, *p*-coumaric and vanillic acids, and lutein and lignans) and vitamin E previously identified in olive [105]. 

Phenolic compounds extracted from olive oil processing waste have also been used in combination with other plant extracts rich in PCs—for instance, chestnut. In this regard, Hoseinifar et al. [106] evaluated the effect of dietary supplementation of a mixture of PCs extracted from olive mill wastewater (OMWW) and chestnut wood (CW) (9:1, OMWW:CW) in concentrations of 0.5, 1.0, and 2.0 g/kg of feed, on the innate immune response of convict cichlid (*Amatitlania nigrofasciata*). The authors reported that mucus total protein levels and lysozyme activities increased in fish fed OMWW:CW. Besides, they indicated that the supplementation of 2.0 g/kg of OMWW:CW increased serum total protein and total Ig levels, as well as peroxidase and radical scavenging activities. The effect of PCs extracted from OMWW:CW (0.5, 0.1, and 2.0 g/kg) on growth performance and innate immune response also have been evaluated in common carp (*C. carpio* L.) [107]. In this study authors reported that supplementation with OMWW:CW increased skin mucus total proteins and Ig levels and lysozyme, peroxidase, and radical scavenging activities in this species. Serum total Ig levels increased in fish fed 0.1 y 0.2 g/kg of OMWW:CW. Besides, PCs from OMWW:CW improved growth (WG and SGR) and feed utilization (FCR) in common carp. In the study, the improve in the immune response in convict cichlid was attributed to PCs such as hydroxytyrosol, tyrosol, and oleuropein, which have been reported in olive, and to tannins identified in CW [108]. However, the mechanism of action by which PCs exert their immunomodulatory effects is still unknown, so efforts should be directed towards that study area. Further studies are necessary with challenges with infectious bacterial or viral diseases to assess the potential of olive waste and CW byproducts as functional feed additives for the aquaculture.

Terpenes, mainly essential oils, have also been studied for their promoter effect of the immune response. Recently it has been reported that the use of essential oils to enhance the immune system response in aquaculture species is a potential alternative to the use of antibiotics [109]. Peels obtained from citrus processing are a good option as a source of essential oil to use them as additives in aquaculture foods [110]. In this context, Acar et al. [111] evaluated the dietary effect of essential oils obtained from orange peel (*Citrus sinensis*) on the growth of Mozambique tilapia (*Oreochromis mossambicus*) and its resistance against the pathogen *Streptococcus iniae*. The fish were fed a control diet, which does not contain essential oils, and three experimental diets (1, 3, and 5 g/kg) for 12 weeks, after which time the fish were challenged by infection with *S. iniae.* Fish fed essential oils, increased lysozyme, and MPO activities. Besides, the addition of essential oils in 1, 3, and 5 g/kg increased fish survival by 48.33%, 46.67%, and 58.33%, respectively. Limonene, a phenolic monoterpene present in the orange peel essential oil, has antibacterial properties and could be responsible for these effects. In general, the results of this study demonstrated that the inclusion of essential oils in diets for tilapia improves the immune response of the fish and therefore may have the potential to be used as antibiotic substitutes. Furthermore, Baba et al. [112] evaluated the effect of essential oils obtained from the lemon peel (*Citrus limon*) on the immune system and resistance against *Edwardsiella tard* in Mozambique tilapia. Fish fed 5 g/kg and 7.5 g/kg of lemon peel essential oil increased the levels of the immuno-hematological parameters, such as the NBT, the number of white cells, hematocrit, and the activity of lysozyme and MPO. After the feeding trial the organisms were subjected to infection by *E. tard*. Fish fed the control diet showed 80% of mortality percentage, while those fed 5, 7.5, and 10 g/kg of essential oils reduced this percentage to 36.6%, 51.6%, and 58.3%, respectively. All the parameters evaluated in this study are important to determine the immune response of the fish. Particularly, lysozyme and MPO have an important role in the elimination of pathogenic microorganisms. Limonene is the main component of the essential oils of lemon peel (54.4%). This compound exerts antimicrobial activity due to its ability to destabilize the bacterial cell membrane [113]. The authors suggest that the addition of 5 g/kg of essential oil of lemon peels into the diet, exerts an immunostimulatory effect, and increases the resistance of Mozambique tilapia against pathogenic bacteria. From these results it can be concluded that citrus peel essential oils are a natural and safe alternative for the formulation of food for aquaculture species.

β-glucans have been widely studied as immunostimulants for organisms of interest in aquaculture, such as shrimp and Nile tilapia [114,115]. β-glucans evaluated are usually obtained from fungi or yeasts [31]. Nevertheless, Chirapongsatonkul et al. [116] recently evaluated the potential of β-glucans obtained from split gill mushroom (*Schizophyllum commune*) cultivation waste as immunostimulant. The authors obtained a crude glucan extract from mycelium containing spent mushroom substrate (SMS) of the *S. commune* to stimulate immune system of Nile tilapia. Fish were injected with 100 µg/mL of glucan extract and after six hours they were challenged with *Aeromonas veronii*. Fish treated with the crude glucan extract showed an increase in immune parameters (Ig, lysozyme) and up-regulation in the expression of cytokine genes (TNF-α, IL-1β, and NF-κB) related to the immune response. Glucan extract treatment also increased the survival rate of Nile tilapia infected with *A. veronii*. These results demonstrate the possibility of use of crude glucan extracts from mushroom cultivation waste to improve the immune response in tilapia. 

Cereals, such as oats, barley, and wheat are a rich source of β-glucans [117] and might be used as additives in diets for fish. For instance, Udayangani et al. [118] evaluated the effect of β-glucans from the endosperm of oat grains on the immune response of zebrafish larvae against *E. tard*. Once the larvae hatched, they were kept in solution for three days with two concentrations of β-glucan (100 and 500 µg/mL). Afterward, the larvae were exposed to the pathogen *E. tard* and the expression of cytokines, lysozyme, and survival percentage were determined. Treatment with 500 µg/mL of β-glucan significantly increased the up-regulation of the expression of lysozyme and cytokine genes (TNF-α, IL-1β, IL-10, and IL-12) related to the immune response, as well as the survival rate. Therefore, β-glucans have potential in the aquaculture industry as promoters of the immune system for larval stages of fish. However, waste from cereal processing has not yet been exploited to obtain β-glucans for the development of aquaculture feeds with immunostimulant effects. 

### 4.3. Bioactive Compounds as Modulators of the Intestinal Microbiota

The modulation of the intestinal microbiota through the use of prebiotics (inulin, galactooligosaccharides, and xylooligosaccharides) has received significant attention due to the growing need to (i) replace probiotics due to their high purchasing value, (ii) reduce the incidence of infectious diseases, (iii) improve the health status of aquaculture organisms, and (iv) increase crop production and profitability [78,119].

Inulin is the most widely used prebiotic in aquaculture [78]. The use of this prebiotic increases the lactic acid bacteria population in the gut of surubies (*Pseudoplatystoma* sp.) and beluga sturgeon (*Spindle spindle*) and decreases *Vibrio* spp. in turbot (*Psseta maxima*). These effects are related to inulin inclusion doses [120,121,122], so further research is still required in this context.

Although there are few studies on the use of prebiotics obtained from agro-industrial wastes, some research has been conducted on prebiotics extracted from cereals, such as wheat. In this regard, Geraylou et al. [123] evaluated the effect of dietary inclusion of wheat bran arabinoxylans (20 g/kg and 40 g/kg), on the composition of the intestinal microbiota of Siberian sturgeon (*Acipenser baerii*). The authors reported that fish fed 20 g/kg and 40 g/kg of arabinoxylans showed an increase in the relative abundance of Eubacteriaceae, Clostridiaceae, Streptococcaceae, and Lactobacillaceae and in Bacillaceae, respectively. Besides, dietary inclusion of wheat bran arabinoxylan oligosaccharides (20 g/kg) modulated the growth of *Lactococcus* sp., *Lactobacillus* sp., *E. budayi*, and several species of the genus *Clostridium*. Furthermore, it has been reported that arabinoxylan oligosaccharides form wheat bran suppress the growth of *Aeromonas* sp., *Citrobacter freundii*, and *Escherichia coli* and increase the content of short-chain fatty acids (acetate and butyrate) in the intestine of Siberian sturgeon [124]. In both studies, the increase in the abundance of the mentioned bacteria is because they possess enzymes (endo-1,4-β-xylanases, α-L-arabinofuranosidases, β-xylosidases, α-glucuronidases, and feruloyl esterases) with capacity to ferment the prebiotics evaluated [125]. It is concluded that wheat bran prebiotics have an impact on the composition of the intestinal microbiota and that the increase in the abundance of lactic acid bacteria and short-chain fatty acids could provide health benefits of Siberian sturgeon.

On the other hand, the use of PCs as prebiotics is of recent interest, so aquaculture studies are scarce. In this context, the effect of PCs of the OMWW on the gut microbiota of narrow clawed crayfish (*Astacus leptodactylus*) was evaluated [126]. Supplementation of 0.5 and 5 g/kg significantly reduced the total intestinal microbiota, with the exclusion of anaerobes and yeasts. This might be because some bacterial groups use PCs as susbtrate, such as lactobacilli. Several PCs present in OMWW, such as, hydroxytyrosol and tyrosol exert antimicrobial properties against bacteria responsible for intestinal infections [127]. The lack of information on the use of PCs extracted from plant residues is a field worth exploring, especially since it is currently known that PCs provide beneficial effects on human health through the increase of bacterial populations beneficial and short-chain fatty acid content [79].

A summary of recent studies in which bioactive compounds obtained from agro-industrial waste were used as feed additives or vaccines, as well as their in vivo effect on antioxidant status, immune system, and microbiota is shown in Table 1.

## 5. Conclusions

There is little research aimed to the valorization of agro-industrial waste and its use as a source of bioactive compounds to incorporate them into aquaculture food. Above all, to characterize these wastes in their nutritional aspects, as well as in the quantity and type of bioactive that they present, it is of utmost importance. The activity of bioactive compounds, such as phenolic compounds, terpenes and β-glucans, depends on their chemical structure, the source, the doses, and if it is isolated or in presence of other compounds, as well as the species used as study model. Therefore, these compounds are required to be evaluated in different aquatic organisms of commercial interest, such as shrimp, tilapia, white snook, and snapper, among others, to determine their biological effect, whether antioxidant, immunostimulant, or microbiota modulator. The above reveals the great window of opportunity that exists to explore this topic.

## Figures and Tables

**Figure 1 foods-09-00843-f001:**
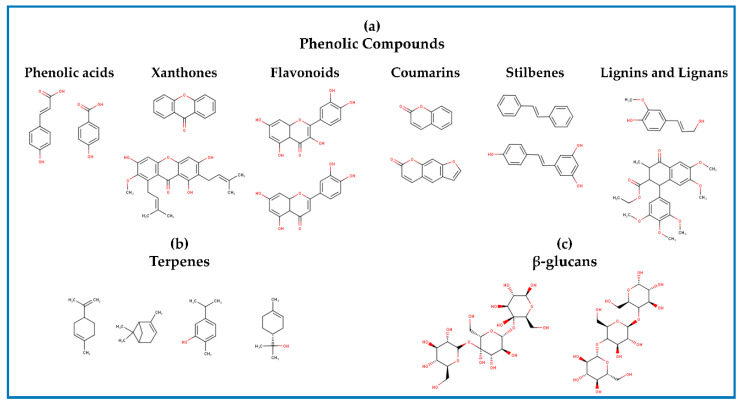
Chemical structure of (**a**) phenolic compounds, (**b**) terpenes, and (**c**) β-glucans.

**Figure 2 foods-09-00843-f002:**
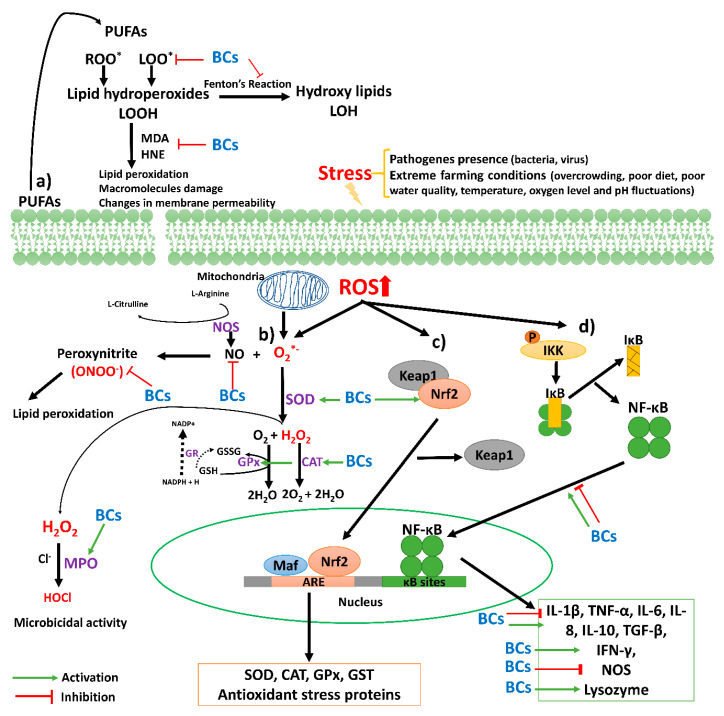
Graphical representation of the mechanism of action of bioactive compounds on the antioxidant and immune response. (**a**) Lipid peroxidation chain reaction, (**b**) antioxidant enzymes reaction, (**c**) Nrf2 pathway associated to the antioxidant response, and (**d**) NF-κB pathway associated to the immune response. Abbreviations: ARE—antioxidant response element; BCs—bioactive compounds; CAT—catalase; GPx—glutathione peroxidase; GR—glutathione reductase; GSH—glutathione; GSSG—oxidized glutathione; GST—glutathione transferase; HNE—4-hydroxynonenal; HOCl—hypochlorous acid; IFN-γ—interferon-gamma; IkB—inhibitor protein of nuclear factor kappa-light chain-enhancer of activated B cells; IKK—kinase complex; IL—interleukin; Keap1—Kelch-like ECH-associated protein 1; LOO*—lipid hydroperoxyl radical; Maf—musculoaponeurotic fibrosarcoma; MDA—malondialdehyde; MPO—myeloperoxidase; NADP+—nicotinamide adenine dinucleotide phosphate; NADPH—reduced form of NADP; NF-κB—nuclear factor kappa-light chain-enhancer of activated B cells; NOS—nitric oxide synthase; Nrf2—NF-E2-related factor 2; PUFAs—polyunsaturated fatty acids; ROO*—peroxyl radical; SOD—superoxide dismutase; TGF-β—transforming growth factor-beta; TNF-α—tumor necrosis factor-alpha.

**Figure 3 foods-09-00843-f003:**
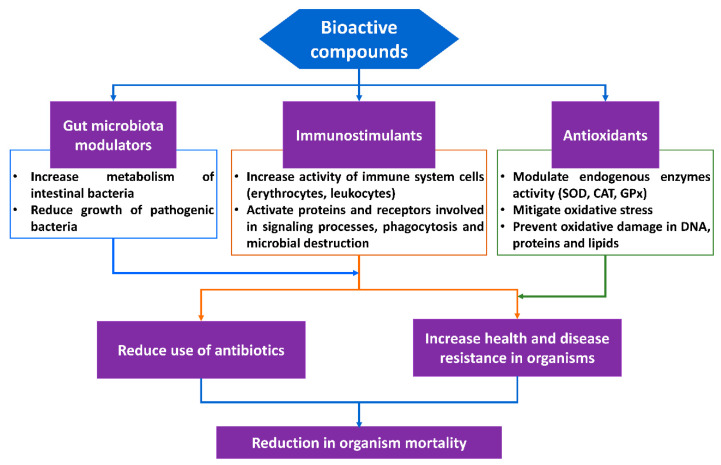
Biological properties demonstrated for bioactive compounds used as food additives for aquatic organisms.

**Table 1 foods-09-00843-t001:** Bioactive compounds from agro-industrial waste and biological effect on aquatic organisms.

Bioactive Compound.	Source (Doses)	Species	Biological Properties	Effect
Phenolic compounds	Corn silk extract (3.5 g/kg)	*Oreochromis niloticus*	Antioxidant, Immunostimulant	↓MDA ↑Lysozyme, against *Aeromonas hydrophila* [89]
Phenolic compounds	Sorghum distillery residue (200 g/kg)	*Mugil cephalus*	Antioxidant	↓LDL oxidation ↑TAC [92]
Phenolic compounds	Banana peel (10, 30, 50, 70 g/kg)	*Labeo rohita*	Antioxidant, Immunostimulant	↓MDA ↑SOD, CAT, GPx ↑IL-1 β, TNF- α ↑Survival rate against *Aeromonas hydrophila* [94]
Phenolic compounds	Orange peel (2, 4, 6, 8 g/kg)	*O. Oreochromis niloticus*	Antioxidant	↑SOD, CAT, GPx [95]
Phenolic compounds: gallic, 2-hydroxycinnamic and protocatechuic acids, quercetin, mangiferine, methyl gallate, ethyl gallate	Mango peel extract (50, 100, 150, 200 mg/kg)	*Danio rerio*	Antioxidant	↑CAT ↓MDA [98]
β-glucans	Mushroom stalk waste extract (5, 10 g/kg)	*Oreochromis niloticus*	Antioxidant	↑SOD, CAT [99,100]
Phenolic compounds: proanthocyanidin, catechins, epicatechins	Grape seeds (0.1, 0.2 g/kg)	*Dicentrarchus labrax* L.	Immunostimulant	↓IL-1β, IL-6. ↑IFN-γ ↑MMCs [75]
Phenolic compounds: catechins, epigallocatechins	Grape seeds (0.1, 0.2 g/kg)	*Dicentrarchus labrax* L.	Immunostimulant	↑Peroxidase ↑Dopa-oxidase ↑MMCs [103]
Phenolic compounds	Olive waste cake (0.5, 2.5, 5 g/kg)	*Oncorhynchus mykiss*	Antioxidant, Immunostimulant	↑SOD, GPx ↑Lysozyme, Ig ↑IL-8 ↓TGF-β [104]
Phenolic compounds	Mixture of chestnut wood and olive mill wastewater extract (0.5, 1, 2 g/kg)	*Amatitlania nigrofasciata*	Antioxidant, Immunostimulant	↑Growth performance ↑Ig ↑Lysozyme, peroxidase ↑CAT [106]
Phenolic compounds	Mixture of chestnut wood and olive mill wastewater extract (0.5, 1, 2 g/kg)	*Cyprinus carpio* L.	Immunostimulant	↑Growth performance ↑Ig, lysozyme ↑CAT, peroxidase [107]
Essential oils	Orange peel (1, 3, 5 g/kg)	*Oreochromis mossambicus*	Immunostimulant	↑Lysozyme, MPO ↑Survival rate against *Streptococcus iniae* [111]
Essential oils	Lemon peel (5, 7.5, 10 g/kg)	*Oreochromis mossambicus*	Immunostimulant	↑NBT ↑Lysozyme, MPO ↓Mortality against *Edwardsiella tarda* [112]
Essential oils	Lemon peel (10, 20, 50, 80 g/kg)	*Labeo victorianus*	Immunostimulant	↑Red blood cells, leucocytes, hematocrits, neutrophils ↑Ig, lysozyme ↓Mortality against *Aeromonas hydrophila* [128]
Glucans	Split gill mushroom cultivation waste extract (100 µg/mL)	*Oreochromis niloticus*	Immunostimulant	↑Ig, lysozyme ↑TNF-α, IL-1β, NF-κB ↑Survival rate against *Aeronomas veronii* [116]
Phenolic compounds	Olive mill waste water (0.5, 5 g/kg)	*Astacus leptodactylus*	Antioxidant, Immunostimulant, Microbiota modulation	↑Growth performance ↑CAT, GR ↑Haemocytes ↓Total intestinal bacteria [126]
Arabinoxylans oligosaccharides	Wheat bran (20, 40 g/kg)	*Acipenser baerii*	Microbiota modulation	↑Eubacteriaceae, Clostridiaceae, Streptococcaceae, Lactobacillaceae, Bacillaceae [123]
Arabinoxylans oligosaccharides	Wheat bran (20 g/kg)	*Acipenser baerii*	Immunostimulant, Microbiota modulation	↑Peroxidase, phagocytic activity ↓*Aeromonas* sp., *Citrobacter freundii*, *Escherichia coli* ↑Short-chain fatty acids [124]

CAT—catalase; GPx—glutathione peroxidase; GR—glutathione reductase; IFN-γ—interferon-gamma; Ig—immunoglobulins; IL—interleukin; LDL—low-density lipoproteins; MDA—malondialdehyde; MMCs—melanomacrophage centers; MPO—myeloperoxidase; NBT—nitroblue tetrazolium; NF-κB—nuclear factor kappa light-chain-enhancer of activated B cells; SOD—superoxide dismutase; TAC—total antioxidant capacity; TGF-β—transforming growth factor-beta; TNF-α—tumor necrosis factor-alpha.

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
