# Peer review of "Exploitation of Agro-Industrial Waste as Potential Source of Bioactive Compounds for Aquaculture"

_foods, 2020, doi:10.3390/foods9070843_

Round 1

Reviewer 1 Report

Dear Authors 

It was exceptional to read a very well managed review paper of yours. Agro-waste management is not a new matter but it is truly one of the most important issues in the agriculture industry of the developing countries which is also the best way to convert great loss to a greater benefit. 

I have minor requests for your paper to be considered.

Figure 1: If you could use Chemical structure drawing software (they are free of charge) to draw it by yourself then it will add a positive point to your presentation.

Scheme 1: Please make it in a higher resolution and use darker colors in general to improve the illustration.

Table 1: Please do not use abbreviations for the "Bioactive compound" column as the first one and refrain from using the abbreviated form of "species" in the related column. In case of the lack of space, you can abbreviate the Biological properties" column because words are too repetitive, also it is possible if you merge "effect" and "reference" column (just add the [reference] at the end of each effect).

Author Response

COVER LETTER

Anonymous Reviewer

Foods-Special Issue: "Recovery of High Value-Added Compounds from Food By-Product"

Dear Reviewer

Through this letter I allow myself to submit the response to your comments regarding the manuscript foods-830384. I would like to thank you for your valuable observations. Certainly, the comments and suggestions made by you will help to improve this manuscript and to enhance its relevance to the scientific community.

Below you will find the answer to each of your comments. I hope that all your observations were amended correctly and as expected.

Reviewer’s comments

It was exceptional to read a very well managed review paper of yours. Agro-waste management is not a new matter but it is truly one of the most important issues in the agriculture industry of the developing countries which is also the best way to convert great loss to a greater benefit. I have minor requests for your paper to be considered.

Figure 1: If you could use Chemical structure drawing software (they are free of charge) to draw it by yourself then it will add a positive point to your presentation.

Response: The Figure 1 was substituted with a new version (line 91). We used the free-version of the software MarvinSketch by ChemAxon to draw and create all the chemical structures included.

Scheme 1: Please make it in a higher resolution and use darker colors in general to improve the illustration.

Response: The Scheme 1 was substituted with a new version (line 213). The colors were changed, as well as the font style, in order to improve the definition. The new file (.png) corresponding Scheme 1 presents a resolution of 12500x8000 pixels width/height.

Table 1: Please do not use abbreviations for the "Bioactive compound" column as the first one and refrain from using the abbreviated form of "species" in the related column. In case of the lack of space, you can abbreviate the Biological properties" column because words are too repetitive, also it is possible if you merge "effect" and "reference" column (just add the [reference] at the end of each effect).

Response: As suggested, the table 1 (line 431) was modified. The abbreviation PCs was substituted for “phenolic compounds”. The taxonomic names of the organisms now appear complete, whit no abbreviations, in the column “Species”. Besides, the columns “Effect” and “Reference were merged in one column with the name “Effect”.

I hope the modifications to the manuscript are in your satisfaction.

Sincerely,

Dra. Crisantema Hernández

Mazatlán, Sinaloa, México, June 15th, 2020

Reviewer 2 Report

This is a review article that refers readers to other edited sources. Whereas for a beginner this may a serve as a guide, there is no extensive experimental description, numerical values and absolutely no equations (other than the three bunch of classes) appear in the manuscript.   A considerable degree of scientific aditing is required.

To be more detailed:

The cited references were very good and inclusive.
However, (with the exception of the three general classes) there were no formulae, equations, mechanisms included.
The reader, according to the authors is referred to other references in trying to justify the concepts around which the overall topic of the paper is revolved.
Therefore I would have liked to see more equations, more extensive experimental description before conclusions are made as well as the mechanistic pathways of e.g., intermediates formed.

Author Response

COVER LETTER

Anonymous Reviewer

Foods-Special Issue: "Recovery of High Value-Added Compounds from Food By-Product"

Dear Reviewer 2

Through this letter I allow myself to submit the response to your comments regarding the manuscript foods-830384. I would like to thank you for your valuable observations. Certainly, the comments and suggestions made by you will help to improve this manuscript and to enhance its relevance to the scientific community.

Below you will find the answer to each of your comments. I hope that all your observations were amended correctly and as expected.

Reviewer’s comments

This is a review article that refers readers to other edited sources. Whereas for a beginner this may a serve as a guide, there is no extensive experimental description, numerical values and absolutely no equations (other than the three bunch of classes) appear in the manuscript. A considerable degree of scientific aditing is required.

To be more detailed:

  • The cited references were very good and inclusive.
  • However, (with the exception of the three general classes) there were no formulae, equations, mechanisms included.

Response: Equations and formulae related to the mechanistic pathways of antioxidant and immune response in organisms, as well as mode of action of bioactive compounds, were added in a new figure (line 189, Fig. 2), expecting to satisfy the reviewer’s observation.

  • The reader, according to the authors is referred to other references in trying to justify the concepts around which the overall topic of the paper is revolved. Therefore, I would have liked to see more equations, more extensive experimental description before conclusions are made as well as the mechanistic pathways of e.g., intermediates formed.

Response: As suggested, the mechanisms of action of bioactive compounds were better explained in the sections 3.1 (line 150-156, line 165-183), 3.2 (line 234-257) and 3.3 (line 269-275, line 279-298). Furthermore, a graphical representation of the mechanistic pathways was added to the review, in order to better explain the effects of bioactive compounds (line 189, Figure 2).

I hope the modifications and information addition to the manuscript are in your satisfaction.

Sincerely,

Dra. Crisantema Hernández

Mazatlán, Sinaloa, México, June 15th, 2020

Reviewer 3 Report

The Review is focused on the agro-industrial waste as materials that may be potentially useful as sources of bioactive compounds for aquaculture organisms. Paper is prepared properly and is supported by numerous literature references. Authors paid attention to the selected groups of bioactive compounds, their biological properties and final attention is focused on their application in aquaculture. Proposed work is worth noticing, but some minor improvements are suggested. All details are provided below.

  • Paragraph concerning the explanation of the term “bioactivity” needs to be added to the Introduction of the manuscript.
  • In section 2.1. of the article it was reported by Authors that phenolic compounds can exert the biological activity by interacting with different components. Please, comment this aspect more widely.
  • The order of two sections of the Review should be changed, i.e. section 2. and section 3. Please, place firstly this part concerning the description and the meaningless of particular bioactive properties and next section which contains the characterization of selected bioactive compounds including previously described properties.
  • Section Conclusions should be improved to be more specific.
  • Paper contains some grammar mistakes (e.g. “these bioactive compounds as food additives for aquaculture has been addressed” instead of “these bioactive compounds as food additives for aquaculture have been addressed”) that should be corrected.

Author Response

COVER LETTER

Anonymous Reviewer

Foods-Special Issue: "Recovery of High Value-Added Compounds from Food By-Product"

Dear Reviewer 3

Through this letter I allow myself to submit the response to your comments regarding the manuscript foods-830384. I would like to thank you for your valuable observations. Certainly, the comments and suggestions made by you will help to improve this manuscript and to enhance its relevance to the scientific community.

Below you will find the answer to each of your comments. I hope that all your observations were amended correctly and as expected.

Reviewer’s comments

The Review is focused on the agro-industrial waste as materials that may be potentially useful as sources of bioactive compounds for aquaculture organisms. Paper is prepared properly and is supported by numerous literature references. Authors paid attention to the selected groups of bioactive compounds, their biological properties and final attention is focused on their application in aquaculture. Proposed work is worth noticing, but some minor improvements are suggested. All details are provided below.

  • Paragraph concerning the explanation of the term “bioactivity” needs to be added to the Introduction of the manuscript.

Response:  As requested, a brief definition of the term “bioactivity” was added in the introduction (Line 38-42).

  • In section 2.1. of the article it was reported by Authors that phenolic compounds can exert the biological activity by interacting with different components. Please, comment this aspect more widely.

Response: Further information regarding the mechanisms by which phenolic compounds exert their biological activity was added in lines 81-87.

  • The order of two sections of the Review should be changed, i.e. section 2. and section 3. Please, place firstly this part concerning the description and the meaningless of particular bioactive properties and next section which contains the characterization of selected bioactive compounds including previously described properties.

Response: In section 2 we included a brief description of the compounds to introduce the reader in the topic of bioactive compounds and the different chemical structures that they possess. The chemical structure is later related to the biological activity, in section 3. Since in section 3, the chemical structure is being associated to each biological we thought that it is important that we describe, previously, the structural conformation of the bioactive compounds. Therefore, even though this is a very valuable suggestion it was not possible to amend it exactly as the reviewer suggested. 

  • Section Conclusions should be improved to be more specific. Response: The aim of this review is to address the potential to exploit agro-industrial waste to obtain bioactive compounds for the development of functional foods for aquaculture, therefore, we propose a conclusion specific for this objective, accordingly with the suggestion made by the reviewer.

  • Paper contains some grammar mistakes (e.g. “these bioactive compounds as food additives for aquaculture has been addressed” instead of “these bioactive compounds as food additives for aquaculture have been addressed”) that should be corrected.

Response: The grammar mistakes have been corrected in lines 21, 76-77, and 127.

I hope the modifications and information addition to the manuscript are in your satisfaction.

Sincerely,

Dra. Crisantema Hernández

Mazatlán, Sinaloa, México, June 15th, 2020

Round 2

Reviewer 2 Report

Glad the authors clarified and fulfilled my suggestions. Please publish as is